# Early Spread of COVID-19 in the Air-Polluted Regions of Eight Severely Affected Countries

Riccardo Pansini [1,2,3] and Davide Fornacca [2,4,*]

1 Behavioral and Experimental Economics Research Center, Statistic and Mathematics College, Yunnan University of Finance and Economics, Kunming 650221, China; r.pansini@gmail.com
2 Institute of Eastern-Himalaya Biodiversity Research, Dali University, Dali 671003, China
3 Department of Economics and Finance, Global Research Unit, City University of Hong Kong, Hong Kong 999077, China
4 Institute for Environmental Sciences, University of Geneva, 1205 Geneva, Switzerland
* Correspondence: fornacca@eastern-himalaya.cn

**Abstract:** COVID-19 escalated into a pandemic posing several humanitarian as well as scientific challenges. We here investigated the geographical character of the early spread of the infection and correlated it with several annual satellite and ground indexes of air quality in China, the United States, Italy, Iran, France, Spain, Germany, and the United Kingdom. The time of the analysis corresponded with the end of the first wave infection in China, namely June 2020. We found more viral infections in those areas afflicted by high PM 2.5 and nitrogen dioxide values. Higher mortality was also correlated with relatively poor air quality. In Italy, the correspondence between the Po Valley pollution and SARS-CoV-2 infections and induced mortality was the starkest, originating right in the most polluted European area. Spain and Germany did not present a noticeable gradient of pollution levels causing non-significant correlations. Densely populated areas were often hotspots of lower air quality levels but were not always correlated with a higher viral incidence. Air pollution has long been recognised as a high risk factor for several respiratory-related diseases and conditions, and it now appears to be a risk factor for COVID-19 as well. As such, air pollution should always be included as a factor for the study of airborne epidemics and further included in public health policies.

**Keywords:** air pollution; COVID-19; coronavirus; virulence; risk factor; satellite air quality

## 1. Introduction

From the first detected outbreak of a new member of the coronavirus (CoV) family [1] in Wuhan, Hubei Province, China [2–4], SARS-CoV-2 [5] has rapidly spread around the world [6], with governments and institutions showing mixed results in its effective containment [7,8]. Certain regions have been much more adversely impacted in terms of infections and mortality rates than others, and the full reasons for this are not yet clear. This paper shows compelling evidence of a correlation between air pollution and incidence of COVID-19 in eight of the first countries known to have experienced an initial fast spread of the virus.

Air pollution is notoriously known to cause health problems and, in particular, respiratory diseases to individuals exposed for longer than several days per year [9–14]. Moreover, pollutants in the air that get absorbed systemically are significant underlying contributors to the emergence of respiratory viral infections [15], including the previous SARS-CoV-1 [16]. Air pollution strongly associates with other respiratory infections [9,12,15,17–24], inducing higher mortalities [10,11]. In particular, airborne particulate matter (PM 2.5 and PM 10) has been linked to respiratory disease hospitalisations for pneumonia and chronic pulmonary diseases [17–23]. The ACE2 receptors of bronchial, alveolar, interstitial, and other pulmonary cells can be involved in chronic cellular inflammation due to air pollution and, concurrently, SARS-CoV-2 [25]. Some further experimental evidence shows that emissions

from diesel and coal affect the lungs, causing pathological immune response and inflammations [26,27], voiding past disputes [28] that only high concentrations of these gasses are needed to cause pathologies.

A number of airborne microorganisms can directly infect other people's mucosae or travel further into the air and onto surfaces, causing delayed infections. The particles of several pollutants such as PMs and $NO_2$ can act as a vector for the spread and extended survival in the air of bioaerosols [29–34], including viruses [35–39] such as measles, the avian flu H5N1, and the syncytial virus. A first hypothesis in this direction, not examined here, has arisen for SARS-CoV-2 in northern Italy [40,41].

The strong containment measures adopted firstly by the Chinese government have necessarily biased the natural virus spread [42,43], not allowing the virus to distribute evenly across the country's territory. Recent findings have shown that non-pharmaceutical interventions such as lockdowns significantly decreased the transmission of the virus in Europe [44]. What shall be noted, though, is that its appearance was recorded in a Chinese area affected by some of the highest air pollutions in the world [45], and it showed a relatively high virulence there [46]. In the case that, as in Italy [47,48], the onset of the infection went undetected for weeks before the outbreaks became apparent, air pollution might have played a more relevant role in exacerbating the virus impacts on human health.

Several risk factors have been implicated with the fast spread of SARS-CoV-2. We enumerate the most relevant ones classified into three groups: (1) environmental risk factors, (2) social factors, and (3) personal factors.

(1) The temperate-climate world latitudes have been identified as the probable areas to be mostly affected by COVID-19 [49] due to limited exposure to UV light in winter. The sole temperature [50,51] or humidity [52] appear to play less of a role [53]. Indeed, other human coronaviruses (HCoV-229E, HCoV-HKU1, HCoV-NL63, and HCoV-OC43) appear between December and April, and they are undetectable in summer months in temperate regions, leading to winter seasonality behaviour. Nevertheless, sufficiently analysing meteorological factors is very complex, although a little less so in conjunction with air pollution and wind [54].

(2) A high population density boosts the virus spread, but taken alone, it should not be a reliable predictor for higher virulence and higher mortality [55]. Another evident predictor variable is transportation. The virus spread to different countries has been attributed to air travellers [7,56–63]. The surrounding areas of transport hubs such as airports and large train stations should witness the appearance of the virus earlier than other less connected zones, increasing its transmission [43,64–67]. In cities, mass gatherings at events can transform into super-spreading events [68,69].

(3) A number of personal risk factors have further been implicated with higher morbidity and mortality rates of COVID-19, including age, male gender, and smoking status. In particular, smoking has been associated with a higher morbidity and mortality of COVID-19 in men than in women, given how airways receptors become hyperactivated from both air pollution and smoking [70–72].

One environmental factor that needs further investigation, even though it was already reviewed [73–75] and initially analysed [76–78], is the role of long-term exposure to air pollution in the spread of COVID-19 and manifested higher morbidity and mortality rates. The very first appearance of this virus cannot be directly correlated with pollution since, similar to the other SARS coronaviruses, SARS-CoV-2 is alleged to have transferred the host from the originating bats to humans [79]. However, it still appeared in a Chinese area harshly affected by climate change and by some of the highest air pollutions in the world, showing from its onset a high virulence.

Long-term or chronic exposure is defined as continuous or repeated contact with a toxic substance over a long period of time (months or years) [80,81], and it can be expressed by annual averaged data [82]. Therefore, we expand upon our very first study of this kind, which we released in early April 2020, looking at this phenomenon in three countries [83]. By controlling for population size and density, here, we investigate whether there is a

correlation between long-term exposure to air pollution and SARS-CoV-2, causing respiratory diseases in second-order level administrations (United States counties equivalent) of eight countries: China, the United States (US), Italy, Iran, France, Spain, Germany, and the United Kingdom (UK). Our hypotheses were as follows: (1) Is there a higher incidence of COVID-19 infections in each country's areas chronically afflicted by poorer air quality? (2) Is there a higher COVID-19 mortality rate in these highly polluted areas?

It must be noted that from the time that these hypotheses and related results were presented to the public with a preprint dated early June 2020 [83,84], several studies with similar hypotheses have been published. These studies investigated different regions and used a variety of approaches. They are cited in the Introduction and Discussion to complement and support our hypotheses.

## 2. Materials and Methods

As briefly introduced above, the present work was performed from March until June 2020, when the first wave of the global pandemic was considered under control in China. We added analyses for seven other countries particularly affected by the virus at that particular time of the pandemic. Italy was the second country to know a rapid contagion spread, especially in its highly industrialised northern region. The third country investigated was the conterminous US, which had the highest number of infections worldwide, yet was still behind in the pandemic curve due to its later arrival as compared to Asia and Europe. Among the countries where the virus spread earlier, we included Iran, which heavily suffers from severe air pollution due to the ubiquitous use of gas methane, refineries, and heavy traffic. France and Spain were selected because of the high COVID-19 figures but more minor air pollution issues than Italy. Lastly, Germany and the UK represented suitable candidates to feed into the analysis because of the relatively reduced lockdown measures adopted [85].

We evaluated the potential correlation between air quality metrics and infections at the finest granularity available. Owing to the differences in the virus advancement stage in each country and the different methodologies employed to record COVID-19 infections and deaths as well as testing policies, the data for each country were analysed separately. We evaluated the potential correlation between air quality metrics and infections at the finest granularity available, controlling both COVID-19 and air pollution variables for potential relationships with population densities as well as the presence of bivariate virus/pollution spatial clusters. Then, the results and differences in the pattern between countries are discussed.

### 2.1. Data Collection and Processing

The COVID-19 datasets were compiled at the second-order administrative subdivision level (US counties equivalent), using the last available information at the time of the analysis (beginning of June); however, a few geographical and time adaptations were required for some contentious administrations that do not make public all the data. In particular, for Iran, we were able to find data of infections only, and at the first-order administrative level only. The Chinese dataset includes the 17 April update with a 50% increase in deaths in Wuhan city [86]. Deaths in Italy were available only at the regional level; therefore, two different datasets were compiled. The autonomous communities of Catalonia, Galicia, and Pais Vasco in Spain provided figures at the first administration level only [87], so we considered them at the same level as provinces. For France, COVID-19 deaths were available only at the department level. Finally, in the UK, the data for Scotland were organised following the National Health Service (NHS) subdivisions rather than the second-order administration scale.

Both infections and deaths due to COVID-19 were collected and normalised by population size per administration unit (100,000 residents), and mortality rates (number of deaths/number of infections × 100) were calculated. Population densities for each unit's area were extracted at 1 square km resolution.

Air quality information was retrieved from long-term satellite observations and averaged at the administrative unit level for each country. The first observations were global annual PM 2.5 grids from MODIS, MISR, and SeaWiFS Aerosol Optical Depth (AOD) with GWR, v1 (1998–2016), and they were obtained from NASA's Socioeconomic Data and Applications Center [88,89]. From the same repository, we retrieved a second dataset consisting of the Global 3-Year Running Mean Ground-Level NO2 Grids from GOME, SCIAMACHY, and GOME-2, v1 (1996–2012) [90,91]. For both products, the annual grids were first reduced to an average multi-year image and, afterwards, the mean of all grid cells covering every administrative unit was calculated.

Additionally, ground measures for the US, China, and Italy were collected from various sources (Table 1). To every administrative unit, we assigned the air quality value from its related station. If more than one point fell within a given unit, the mean was calculated. No ground measures for the other countries were included in our study.

Combining all these measures poses compilation challenges [92]. Satellite data hold several advantages over ground station data, such as regular and continuous data acquisition, quasi-global coverage, and spatially consistent measurement methodologies [93]. On the other hand, ground stations offer actual measures of single pollutants instead of deriving them from spectral information; however, they require more or less arbitrary estimations (such as interpolation) to fill spatial gaps.

**Table 1.** Detailed information on the datasets used for the viral and pollution analyses.

| | Measuring Unit | Time Period | Format | Source (Access Date Same As Time Period) |
|---|---|---|---|---|
| **COVID-19** | | | | |
| China | Infections, Deaths | Until 23 May 2020 | Tabular Prefecture level | DXY—DX Doctor: http://ncov.dxy.cn/ncovh5/view/en_pneumonia Chinese government health commission |
| Italy | Infections, Deaths | Until 22 May 2020 | Tabular Province and region levels | Github repository: https://github.com/pcm-dpc/COVID-19 Dipartimento della Protezione Civile: http://www.protezionecivile.it/ |
| US | Infections, Deaths | Until 21 May 2020 | Tabular County level | The New York Times Github repository: https://github.com/nytimes/covid-19-data |
| Iran | Infections | Until 22 Mar 2020 | Tabular Province level | IRNA–The Islamic Republic News Agency: https://en.irna.ir/photo/83723991/Iran-s-coronavirus-toll-update-March-22-2020 |
| France | Deaths | Until 22 May 2020 | Tabular Department level | Open Data Platform of the French Government: https://www.data.gouv.fr/fr/datasets/chiffres-cles-concernant-lepidemie-de-covid19-en-france/#_ |
| Spain | Infections, Deaths | Until 2 May 2020 | Tabular Province level | Data from Spanish Ministry of Health. Github: https://github.com/Secuoyas-Experience/covid-19-es |
| Germany | Infections, Deaths | Until 25 May 2020 | Tabular District level | Robert Koch Institut: https://www.rki.de/EN/Home/homepage_node.html |
| UK | Infections, Deaths | Until ca. 1 June 2020 (infections) 24 May 2020 (deaths) | Tabular LTLA/NHS level | Several government sources: https://coronavirus.data.gov.uk/, https://phw.nhs.wales/, https://www.ons.gov.uk/, https://www.nrscotland.gov.uk/, https://www.health-ni.gov.uk/ |

**Table 1.** *Cont.*

| | Measuring Unit | Time Period | Format | Source (Access Date Same As Time Period) |
|---|---|---|---|---|
| **Population** | | | | |
| China | | Estimates 2017 | Tabular Prefecture level | https://www.citypopulation.de/ Data from Province Governments |
| Italy | No. of residents | 2019 | Tabular Province level | Istat—Italian National Institute of Statistics http://dati.istat.it/ |
| US | | Estimates 2018 | Tabular County level | US Census Bureau (on ESRI ArcGIS): https://www.arcgis.com/home/item.html?id=a00d6 b6149b34ed3b833e10fb72ef47b |
| Iran | | 2016 | Tabular Province level | Statistical Center of Iran: https://www.amar.org.ir/ |
| France | | Estimates 2020 | Tabular Department level | Insee—French National Institute of Statistics: https://www.insee.fr/ |
| Spain | | 2019 | Tabular Province level | INE—Spanish National Institute of Statistics: https://www.ine.es/en/index.htm |
| Germany | | Estimates 2018 | Tabular District level | Database of the Federal Statistic Office: https://www-genesis.destatis.de/ |
| UK | | Estimates 2018 | Tabular LTLA/NHS level | U.K. Office of National Statistics https://www.ons.gov.uk/ |
| **Air Quality (ground measures)** | | | | |
| China PM 2.5, PM 10, $O_3$, $NO_2$, $SO_2$, CO | AQI | 2014 | Tabular GPS points | University of Harvard Dataverse: https://dataverse.harvard.edu Data from http://aqicn.org |
| Italy PM 2.5, PM 10 | $\mu g/m^3$ | Annual 2013-2016 | Tabular Location name | Ambient Air Quality Database, WHO, April 2018 https://www.who.int/airpollution/data/cities/en/ |
| US PM 2.5, PM 10, $O_3$, $NO_2$, $SO_2$, CO | $\mu g/m^3$ ppm, ppb | 2019 | Tabular GPS points | EPA—United States Environmental Protection Agency https://www.epa.gov/outdoor-air-quality-data |
| **Air Quality (satellite)** | | | | |
| PM 2.5 | $\mu g/m^3$ | Annual 1998-2016 | Continuous grid (0.01 arc deg.) | Global Annual PM 2.5 Grids from MODIS, MISR and SeaWiFS Aerosol Optical Depth (AOD) with GWR, v1 https://doi.org/10.7927/H4ZK5DQS |
| $NO_2$ | ppb | 3-year running means (1996-2012) | Continuous grid (0.1 arc deg.) | Global 3-Year Running Mean Ground-Level NO2 Grids from GOME, SCIAMACHY and GOME-2, v1 (1996–2012) https://doi.org/10.7927/H4JW8BTT |

## 2.2. Data Collection and Processing

Exploratory analysis of the variables was conducted with a focus on evaluating the air pollution distributions within each country. Due to the highly skewed distributions of both population-adjusted dependent variables, namely COVID-19 infections/100,000 inhabitants, COVID-19 deaths/100,000 inhabitants, and mortality rates (deaths/infections $\times$ 100), we opted for a non-parametric correlation metric. Kendall tau correlation coefficients were employed for all statistical tests.

Since both virus spread and air pollution dynamics present visible spatially dependent dynamics, we identified potential clusters of adjacent administrations using Local Moran's Bivariate statistic [94,95]. This metric also shows which regions mostly explain the resulting correlations by excluding non-significant regions.

These results are illustrated with thematic maps that better highlight the overlap between air quality and COVID-19 distributions within the eight assessed countries.

## 3. Results

### 3.1. Correlation between Air Pollution Variables and COVID-19 Infections, Deaths, and Mortality Rates

Significant positive correlations between air quality variables and COVID-19 infections, deaths, and mortality rates were found in China, the US, Italy, Iran, France, and the UK, but not entirely in Spain and Germany (Tables 2–4). The strongest correlations were found in Italy, both for infections and deaths, while population size and densities did not explain COVID-19 incidence. In China, population densities showed a similar positive

correlation with the virus infections and deaths than air pollution, while in the US and UK, the population density had a stronger correlation than air pollution variables. In the UK, air pollution showed a fair degree of correlation with deaths and mortality but not with the infections. Despite its small sample size (df = 29), Iran showed a significant correlation with $NO_2$ distribution and no incidence from population variables. The results for Spain and Germany showed different patterns. Differences in air pollution could not explain the spread of COVID-19 and its related deaths in Spain; however, the mortality rate varied with $NO_2$ concentration. Moreover, population size and density were negatively correlated with the virus. In a distinct manner, population density weakly explained COVID-19 infections in Germany, while the distribution of fine particulate matter was in some cases weakly negatively correlated. Among the different pollutants analysed, $O_3$ and $SO_2$ measures from ground stations in China and the United States did not show significant correlations with COVID-19 or were negatively correlated, in contrast with the overall results from the other pollutants.

## 3.2. COVID-19 Distribution, Clusters, and Air Quality Maps

Figure 1 reports comparison maps of COVID-19 distributions with the satellite-based PM 2.5 concentrations for the eight analysed countries. These graphical representations also allow for a rapid assessment of the air pollution pattern in each country (basic descriptive statistics for each pollutant can be found in Appendix A, Table A1). While the PM 2.5 maps are continuous surfaces drawn following the same classification scheme across countries, the COVID-19 infections and deaths maps required ad hoc classification adaptations due to different population profiles and infection dynamics. In China, due to the vast population and an apparently effective policy for the containment of the virus, the number of infections per 100,000 residents was relatively low and highly concentrated in the epicentre of the outbreak (Wuhan and the Hubei province). A visual correlation between the two maps can be perceived, especially between the eastern and western parts of the country, which are also highlighted in the cluster map (Figure 2).

**Table 2.** Correlation coefficients between COVID-19 infections per 100,000 inhabitants and air quality variables. Significant Kendall correlations with *p*-values < 0.05 are shown in bold; blue and red shadings indicate positive and negative correlations, respectively.

| | China | | | US | | | Italy (Provinces) | | | Iran | | | Spain | | | Germany | | | UK | | |
|---|---|---|---|---|---|---|---|---|---|---|---|---|---|---|---|---|---|---|---|---|---|
| | df (N-2) | Tau | P Value | df (N-2) | Tau | P Value | df (N-2) | Tau | P value | df (N-2) | Tau | P Value | df (N-2) | Tau | P Value | df (N-2) | Tau | P Value | df (N-2) | Tau | P Value |
| population | 337 | 0.23 | <0.001 | 3102 | 0.26 | <0.001 | 105 | 0.00 | 0.951 | 29 | −0.15 | 0.250 | 41 | −0.27 | 0.010 | 399 | 0.03 | 0.317 | 362 | 0.18 | <0.001 |
| pop. dens | 337 | 0.32 | <0.001 | 3102 | 0.30 | <0.001 | 105 | 0.12 | 0.078 | 29 | 0.14 | 0.279 | 41 | −0.33 | 0.002 | 399 | 0.10 | 0.002 | 362 | 0.21 | <0.001 |
| PM 2.5 sat | 337 | 0.28 | <0.001 | 3102 | 0.25 | <0.001 | 105 | 0.62 | <0.001 | 29 | 0.24 | 0.061 | 41 | −0.03 | 0.778 | 399 | −0.07 | 0.046 | 362 | −0.03 | 0.386 |
| NO$_2$ sat | 337 | 0.24 | <0.001 | 3101 | 0.22 | <0.001 | 105 | 0.55 | <0.001 | 29 | 0.40 | <0.001 | 41 | 0.08 | 0.470 | 399 | −0.03 | 0.375 | 360 | 0.06 | 0.086 |
| PM 2.5 gr | 302 | 0.15 | <0.001 | 427 | 0.21 | <0.001 | 88 | 0.34 | <0.001 | | | | | | | | | | | | |
| PM 10 gr | 302 | 0.04 | 0.330 | 201 | 0.14 | 0.004 | 99 | 0.11 | 0.096 | | | | | | | | | | | | |
| CO gr | 302 | −0.01 | 0.840 | 156 | 0.18 | 0.001 | | | | | | | | | | | | | | | |
| NO$_2$ gr | 302 | 0.12 | 0.002 | 246 | 0.41 | <0.001 | | | | | | | | | | | | | | | |
| O$_3$ gr | 302 | −0.03 | 0.477 | 749 | 0.03 | 0.238 | | | | | | | | | | | | | | | |
| SO$_2$ gr | 302 | −0.01 | 0.843 | 314 | −0.12 | 0.002 | | | | | | | | | | | | | | | |

**Table 3.** Correlation coefficients between COVID-19 deaths per 100,000 inhabitants and air quality variables. Significant Kendall correlations with *p*-values < 0.05 are shown in bold; blue and red shadings indicate positive and negative correlations, respectively.

| | China | | | US | | | Italy (Regions) | | | France | | | Spain | | | Germany | | | UK | | |
|---|---|---|---|---|---|---|---|---|---|---|---|---|---|---|---|---|---|---|---|---|---|
| | df (N-2) | Tau | P Value | df (N-2) | Tau | P Value | df (N-2) | Tau | P Value | df (N-2) | Tau | P Value | df (N-2) | Tau | P Value | df (N-2) | Tau | P Value | df (N-2) | Tau | P Value |
| population | 337 | 0.17 | <0.001 | 3102 | 0.36 | <0.001 | 19 | 0.01 | 0.976 | 94 | 0.17 | 0.015 | 41 | −0.25 | 0.019 | 399 | 0.03 | 0.409 | 362 | 0.13 | <0.001 |
| pop. dens | 337 | 0.16 | <0.001 | 3102 | 0.36 | <0.001 | 19 | 0.16 | 0.323 | 94 | 0.24 | <0.001 | 41 | −0.40 | <0.001 | 399 | 0.05 | 0.153 | 362 | 0.29 | <0.001 |
| infect 100k | 337 | 0.39 | <0.001 | 3102 | 0.55 | <0.001 | 19 | 0.83 | <0.001 | . | . | . | 41 | 0.81 | <0.001 | 399 | 0.65 | <0.001 | 362 | 0.46 | <0.001 |
| PM 2.5 sat | 337 | 0.18 | <0.001 | 3102 | 0.24 | <0.001 | 19 | 0.60 | <0.001 | 94 | 0.56 | <0.001 | 41 | −0.09 | 0.385 | 399 | −0.04 | 0.241 | 362 | 0.16 | <0.001 |
| NO$_2$ sat | 337 | 0.16 | <0.001 | 3101 | 0.26 | <0.001 | 19 | 0.51 | <0.001 | 94 | 0.57 | <0.001 | 41 | 0.08 | 0.470 | 399 | −0.05 | 0.118 | 360 | 0.23 | <0.001 |
| PM 2.5 gr | 302 | 0.18 | <0.001 | 427 | 0.24 | <0.001 | 17 | 0.22 | 0.183 | | | | | | | | | | | | |
| PM 10 gr | 302 | 0.12 | 0.006 | 201 | 0.18 | <.001 | 19 | 0.00 | 1.00 | | | | | | | | | | | | |
| CO gr | 302 | 0.11 | 0.012 | 156 | 0.20 | <.001 | | | | | | | | | | | | | | | |
| NO$_2$ gr | 302 | 0.12 | 0.005 | 246 | 0.42 | <.001 | | | | | | | | | | | | | | | |
| O$_3$ gr | 302 | −0.02 | 0.585 | 749 | 0.03 | 0.173 | | | | | | | | | | | | | | | |
| SO$_2$ gr | 302 | 0.04 | 0.409 | 314 | −0.08 | 0.028 | | | | | | | | | | | | | | | |

**Table 4.** Correlation coefficients between COVID-19 mortality rates and air quality variables. Significant Kendall correlations with *p*-values < 0.05 are shown in bold; blue and red shadings indicate positive and negative correlations, respectively.

| | China | | | US | | | Italy (Regions) | | | Spain | | | Germany | | | UK | | |
|---|---|---|---|---|---|---|---|---|---|---|---|---|---|---|---|---|---|---|
| | df (N-2) | Tau | *P* value | df (N-2) | Tau | *P* value | df (N-2) | Tau | *P* value | df (N-2) | Tau | *P* value | df (N-2) | Tau | *P* value | df (N-2) | Tau | *P* value |
| PM 2.5 sat | 313 | 0.16 | **<0.001** | 2904 | 0.17 | **<0.001** | 19 | 0.45 | **0.004** | 41 | −0.09 | 0.408 | 399 | 0.00 | 0.987 | 361 | 0.25 | **<0.001** |
| NO$_2$ sat | 313 | 0.14 | **0.001** | 2904 | 0.20 | **<0.001** | 19 | 0.34 | **0.031** | 41 | 0.13 | 0.205 | 399 | −0.07 | **0.047** | 360 | 0.20 | **<0.001** |
| PM 2.5 gr | 285 | 0.18 | **<0.001** | 418 | 0.18 | **<0.001** | 17 | 0.07 | 0.674 | | | | | | | | | |
| PM 10 gr | 285 | 0.13 | **0.005** | 191 | 0.15 | **0.002** | 19 | 0.08 | 0.654 | | | | | | | | | |
| CO gr | 285 | 0.12 | **0.007** | 156 | 0.14 | **0.009** | | | | | | | | | | | | |
| NO$_2$ gr | 285 | 0.12 | **0.007** | 239 | 0.26 | **<0.001** | | | | | | | | | | | | |
| O$_3$ gr | 285 | −0.03 | 0.482 | 738 | 0.02 | 0.435 | | | | | | | | | | | | |
| SO$_2$ gr | 285 | 0.06 | 0.178 | 309 | 0.00 | 0.925 | | | | | | | | | | | | |

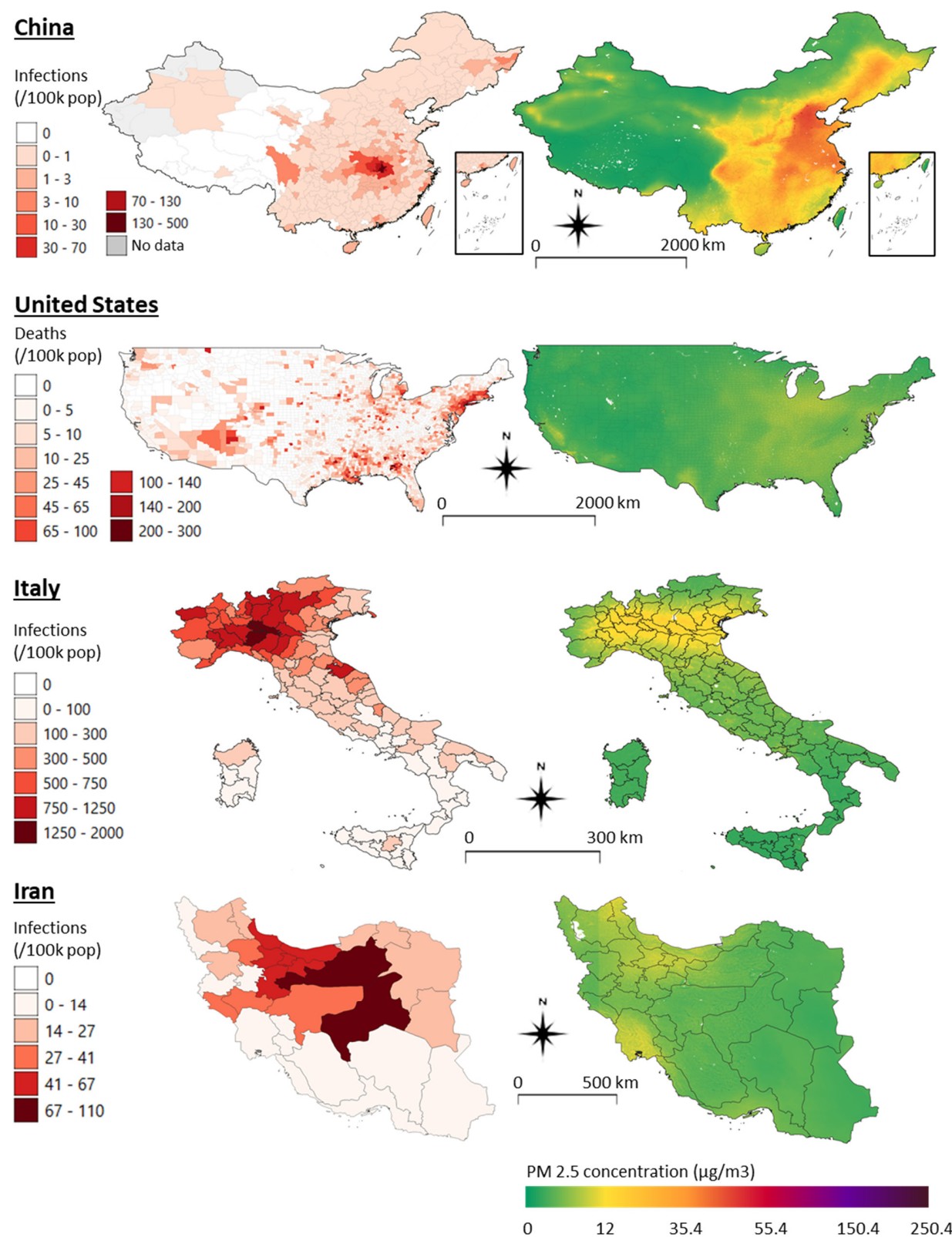

**Figure 1.** *Cont.*

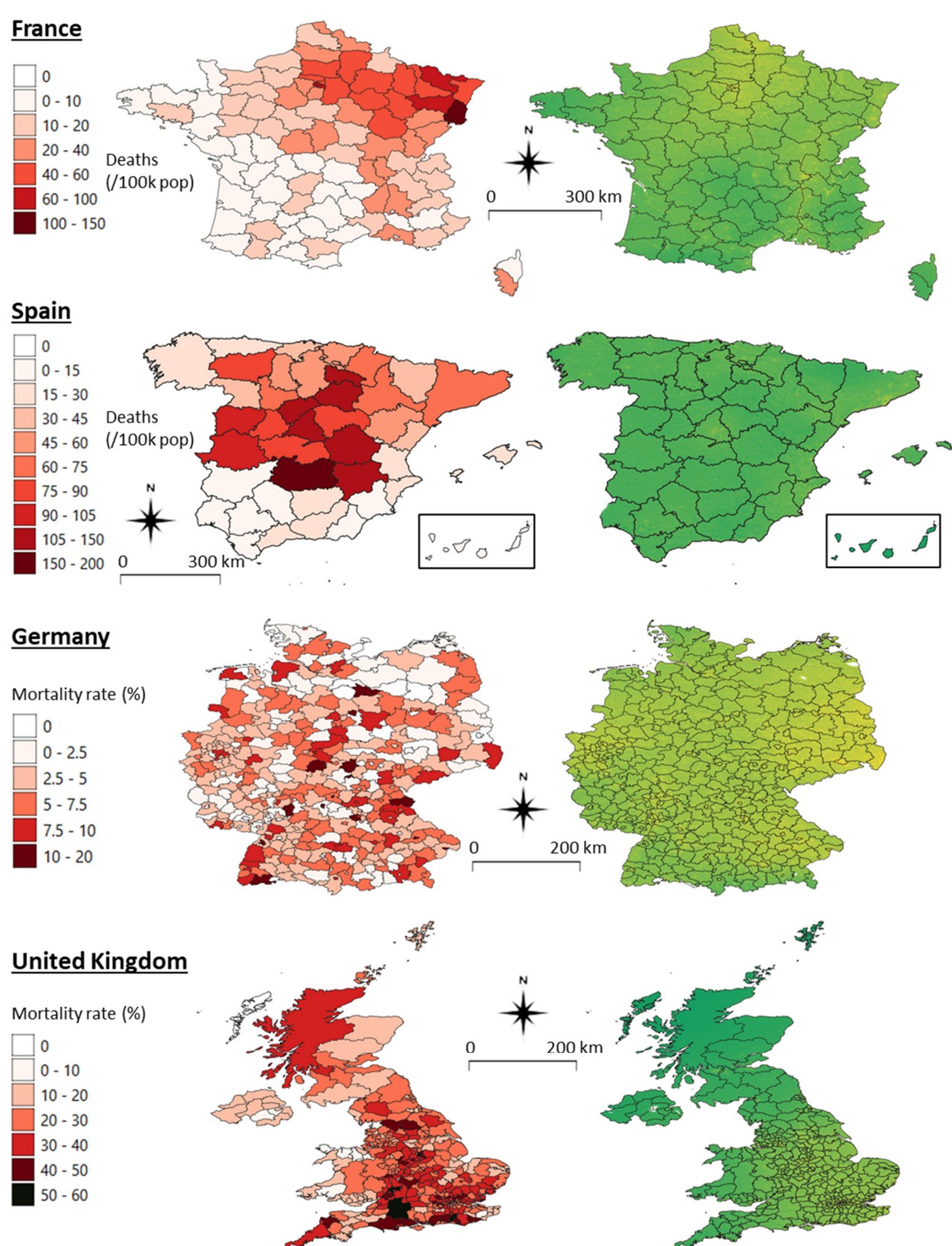

**Figure 1.** Map comparisons of satellite-derived PM 2.5 distributions and COVID-19 infections, deaths (per 100,000 inhabitants), or mortality rates (deaths/infections) in eight countries (see Table 1 for information about dates and data sources). Maps of different countries may not be compared directly due to different classification schemes and spatial scales. Some administrative units' boundaries were adapted according to the COVID-19 data available (e.g., merged districts of Galicia, Catalunia, and Pais Vasco in Spain).

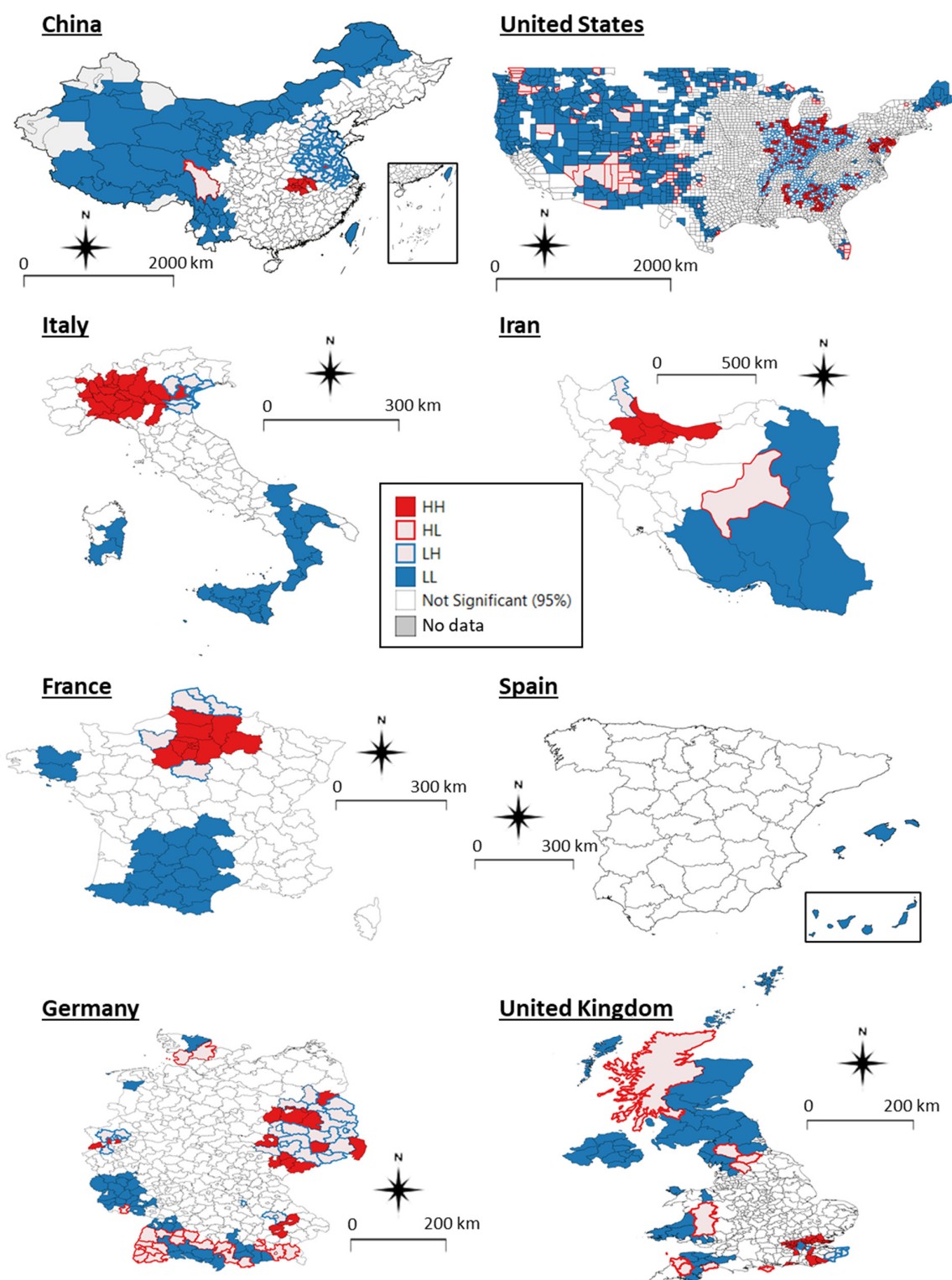

**Figure 2.** Maps of clustered, adjacent administrations resulting from Local Moran Bivariate analysis. Four types of significant spatial relationships exist. HH in filled red: High COVID-19 (infections, deaths, or mortality rates) and High PM 2.5; HL in empty red: High COVID-19 and Low PM 2.5; LH in empty blue: Low COVID-19 and High PM 2.5; LL in filled blue: Low COVID-19 and Low PM 2.5. While HH and LL filled coloured clusters support the COVID-19/air pollution correlation hypothesis, HL and LH empty colours represent spatial outliers in which air pollution does not explain the virus' presence. Note that the COVID-19 variables used in each country are those of Figure 1. Data sources can be found in Table 1.

The presence of outlier clusters (HL and LH) and large non-significant areas in most of the countries partly explains the limited significance and strengths of correlations shown in the tables at the general countries' level. The highly developed and polluted areas in the east of China represent outlier clusters due to the low COVID-19 infections compared to the Hubei province. In the US, the virus noticeably appears to spread over several areas. PM 2.5 differences are not large, but their distribution looks adequately coincident with the deaths. Similar to China, a longitudinal pattern is visible with low-deaths/low-pollution clusters (LL) concentrated in the mid-western part of the country while high-death and high-pollution clusters (HH) are found in the east, along the Mississippi River and the states surrounding New York. However, a high number of outliers of both types (HL and LH) exist. The high correlation results found for Italy are clearly visible. The polluted areas of the Po Valley are those heavily affected by COVID-19 infections. The clusters are clear, and the number of outliers is minimal. While in Iran and France, the correlations are only lightly perceivable, and the cluster maps show a north–south regionalisation pattern similar to Italy. The maps of Spain confirm the absence or weak correlation shown in Tables 2–4, apparently going against our general hypotheses. Nevertheless, PM 2.5 levels in Spain are minimal, as well as their variation—as indicated by the low range and interquartile range (Table A1, Appendix A). The UK map of PM 2.5 shows well the higher concentrations around urban areas and the overall southeastern area where COVID-19 mortality is higher, too. However, also, a few counties/NHS in the north of Scotland are particularly affected by the virus infections, becoming outliers in the clusters map. Finally, COVID-19 mortality in Germany is low, and no apparent distribution pattern can be detected, being quite well spread. Similarly, PM 2.5 concentrations are fairly high all over the country, with peaks in the eastern districts, where a few HH clusters and LH outliers are found. The high number of non-significant clusters and both types of outliers confirm this tendency to a homogenous distribution of COVID-19 and air pollution.

### 3.3. Previous Literature Account

Given the delay between the last of our preprints [84] and the present publication, we include a list of 10 recent studies that support our correlational findings (Table 5). These and other research works are discussed in the next section. It is worth noting that a study by Ogen [96] found a positive correlation between $NO_2$ levels and COVID-19 fatalities in the administrative regions of Spain, Germany, Italy, and France when considered together as a cluster. However, our results showed that within the second-order administrative regions of Germany and Spain, the correlation were not always significant, and it was sometimes negative. Except for our study and these two countries, we were not able to find other works contradicting our initial hypotheses.

**Table 5.** A list of correlational studies between long-term exposure to air pollution and incidence of COVID-19 at a country-wide or cluster of countries level. Studies including smaller geographical areas have not been listed as well as those considering the short-term hypothesis (pollution particles acting as virus carriers). The pollutants are specified whether to having been collected from ground (G) or satellite (S) stations.

| Country | Pollutants (G/S) | Correlation | Comment | References |
|---|---|---|---|---|
| US | PM 2.5 (G) | Positive | Additional cofactors studied | [97,98] |
| Italy | PM 2.5 (G), PM 10 (G), $NO_2$ (G), etc | Positive | Additional cofactors studied | [76,99] |
| Spain, Germany, Italy, France | $NO_2$ (S) | Positive | Differences between countries not considered | [96] |
| Netherlands | PM 2.5 (G) | Positive | Additional cofactors studied | [100,101] |
| Japan | PM 2.5 (G) | Positive | | [102] |
| India | PM 2.5 (G), $NO_2$ (G), $CO_2$ (G) | Positive | | [103] |
| Canada | PM 2.5 (G) | Positive | | [104] |

## 4. Discussion and Conclusions

As a preprint [84], this study was the first to investigate the correlation between COVID-19 and air pollution during the early stage of the pandemic. Specifically, we have assessed long-term air pollution exposure for eight countries, which was measured by satellite and ground sensors as a potential and highly likely risk factor for the incidence of and mortality rates due to SARS-CoV-2. It provides some evidence that the new coronavirus infections are most often found in highly polluted and densely populated areas. In Italy and Iran, air pollution independent from population density explained the distribution pattern of the virus. In addition, in these areas affected by a mixture of air pollutants, the virus killed more frequently than elsewhere. In the questionable case that the figures provided by these eight nations concerning the number of infections and deaths are inaccurate [105], our analyses and conclusions would not need to be reframed. If that were the case, this error would most likely be concentrated in just one or very few administrations, or it would be evenly spread across administrations, not affecting the general significance of the correlations.

In Chinese cities [106] and, more in detail, in the Hubei province, time analyses give preliminary evidence of a correlation between high levels of $NO_2$ and 12-day delayed virus outbreaks [107] and other PM covariates [108,109]. With our paper, we therefore add the long-term exposure effects for China as we did in greater detail before [46]. As shown in the maps, China bears extremely high rates of air pollution, as concentrated in the east. However, COVID-19 infections occurred mainly in the constrained area of Hubei. All evidence suggests that the enforced lockdown was the major factor controlling the virus spread. Nevertheless, it is peculiar that the onset of the pandemic still appeared in one of the most polluted areas of the globe.

In the US, an increase of a mere 1 $\mu g/m^3$ in PM 2.5 was recently found responsible for an 8% higher mortality rate by COVID-19 than baseline from previous years. This is a rate relatively higher than the other 11 demographic co-variables tested [97,98]. Ozone and diesel particulate matter were recently confirmed to be a source of concern over there [110]. With our study, we add PM 10, $NO_2$, and CO measures from ground stations to those analyses.

We found that in Italy, the correspondence between poor air quality and SARS-CoV-2 appearance as well as its induced mortality was the starkest. The area with the largest number of infections and deaths in Italy is the Po Valley, which is also the foremost place of polluted air in Europe [111]. This result was first hypothesised [112] and later confirmed by another study [76] and a remarkable further one [99] that has controlled for five demographic co-variables. The fact that population density does not play a role in the incidence of COVID-19 in Italy and Iran is a result of our investigation that strongly supports the common hypotheses of these other studies and questions the widespread scepticism maintaining that air pollution usually overlaps with areas of high population density, and that the contribution of each to the virus incidence cannot be discerned (tested in Appendix B, Table A2). Other factors to be attributed to such a severe virus incidence in Italy include its very large ageing population, which might have gotten exposed to air pollutants for the longest time. In turn, such pollutants cause other comorbidities [77] and COVID-19 vulnerabilities such as cardiocirculatory diseases.

Unfortunately, information on COVID-19 infections was available for Iran only at the first administration level until 22 March 2020. Nonetheless, the results there are very similar to the Italian ones, with the pollution gradient explaining most of the virus incidence. France provides up-to-date information about deaths only. Despite this limitation, France as well shows highly clustered COVID-19/pollution distributions, resulting in significant positive correlations that confirm our hypotheses.

The absence of correlation found in Spain may be attributable to the high levels of air quality throughout its national territory, which are within the green Air Quality Index standard range, ensuing minimal differences among the provinces. Moreover, the regions most affected by the virus seem to be those less densely populated, which is a

peculiarity still not explained thoroughly in the literature that requires future investigation. In Germany, also, a clear correlation could not be detected because, inversely, pollution is widely spread across its districts. For these two countries, we therefore back up another report [96] that analysed these two countries together with Italy and France at the first-order administrative level. Higher levels of $NO_2$ associated with COVID-19 mortality were found in this super-region.

Finally, in the UK, where containment measures were implemented late compared to other countries, deaths and mortality rates, but not infections alone, are correlated with air pollution, suggesting that when affected by the disease, a weakened respiratory system due to prolonged stress by air pollution increases the risk of mortality in those polluted areas of the southeast.

Despite the significant and consistent correlations of these findings that we collected over three time periods in March [83], April [84] and, as reported here, end of May, their interpretation needs to be cautious. The virus spread in most countries is still ongoing [113] and is being contained [47]. Causation should not be inferred by correlational data alone. Air pollution is just one of the risk factors for increased COVID-19 incidence. It is partly comforting that we find outliers or non-significances via clustering analysis. In fact, the regions flagged as such and prevalent in most countries may become sites for the virus due to other factors than air pollution. It is not necessarily because an area is polluted that it will have a higher frequency of COVID-19. The other external factors involved in SARS-CoV-2 infection include age, pathological comorbidities, access to health care, socioeconomic status, multigenerational housing, travel in crowded transportation hubs, attendance at super-spreading events, etc. In addition, other factors include policies for prevention and containment as well as compliance to measures such as wearing face masks, social distancing, contact tracing, lockdowns, etc.

There are confounding factors, such as how the virus infection was determined in patients by different countries. However, the larger the geographical areas affected by the pandemic, the lower these elements play a role. Finally, it should be noted that by accounting for yearly averaged air quality indexes, we accounted for the long-term exposure to these pollutants, therefore keeping on the conservative side. In fact, these correlations would become even more robust when limiting the analysis to the more polluted winter months, given how they invariably bear lower air quality.

We run these analyses considering eight countries in their second-order administrations' level. If controlling for several other predictors such as demographic variables is something advisable to perform at a single-country level to cross-check for interdependence, including them at an international scale poses an apparent technical limitation [114]. National or federal health systems have different capacities and provide care in distinct ways. In turn, this influences case detections, intensive care capacity, and mortality rates. Cofactors such as the earliest location of the pathogen, population mobility, and patient socioeconomic status or ethnicity may not be accounted reliably between such diverse countries spanning from Asia to the western world, because they are interdependent and only in part nested within countries or administrations, even when included as random factors as in a comprehensive generalised mixed model. Yet, the epidemic, which has turned into a pandemic, might have catered for this limitation; the wider its extent, the more prominent a common factor such as air pollution has become, while other secondary predictors will level out across places.

Left alone, ambient, outdoor air pollution, causing an estimated 4.2 million deaths yearly worldwide [81], is a risk cofactor to be hypothesised in connection with a new respiratory disease, without necessarily having to analyse some of the other cofactors, especially in the temperate climate zones where new countries keep reporting similar correlations between PM 2.5 and the virus: the Netherlands [100,101], controlling for some other medical risk factors; Japan [102], finding a positive correlation in the elderly; and also India, which holds a similar trend in relation to the long-term hypothesis [103] and in relation to the short-term one, too [115,116]. Lastly, Canada [104], Peru [117], with other

Latin America countries plus the Caribbean [118], and Malaysia [119] were reported to bear positive associations. To note that, in the most comprehensive analysis performed during the first infection wave in 126 countries, $CO_2$ and SO emissions correlated with COVID-19 when analysed using the "Our World in Data" database [120].

Since there is now some first evidence that the cross of the virus from animals to humans may have happened earlier than the end of 2019 [79] and further south than in the Chinese city of Wuhan [121], we can speculate that air pollution could have allowed the new epidemic to become recognised due to an influx of patients with weak respiratory systems showing higher morbidity and mortality than influenza. The same seems to have happened in Europe, as the virus, in February 2020, quickly moved from central Europe to the most polluted region of the continent, in northern Italy [47,122].

Further research in the field of genetics will ascertain whether virulence has evolved in the areas of those countries where a gradient of air pollution is present. The initial location of the pathogen, long-distance travelling, and super-spreader events are deemed to be the foremost factors governing the epidemics. Later on, other factors such as hospital capacities, population confinement, and possibly also indoor air pollution may become major predictors for severe infections to keep on manifesting. In relation to short-term exposure to peaks of low air quality levels, the capacity of pollutants to act as viral vectors should be investigated further [123,124]. In fact, particulate matter does act as a medium for the aerial transport of SARS-CoV-2 [40,125,126]. Aggregates of particulate matter with this virus have been collected in the worst affected northern Italian city of Bergamo [127]. If the viral load carried by the aggregates is enough to cause morbidity, pollution would directly act as a vector, broadening the harm done by the human-to-human contagions.

To conclude, these findings are sufficiently significant to prompt researchers studying the public health of industrialised countries to always consider air pollution as a contributing risk factor for COVID-19 and for any other airborne viral epidemics [128]. To overcome the limitations of our study, longitudinal screenings performed on patients from retrospective cohorts may further point at air pollution as a cofactor [129]. These results inform epidemiologists and policymakers on how to prevent future, more frequent and lethal viral outbreaks by curbing air pollution and, ultimately, meeting climate goals [130]. Can the fossil fuel economy carry on unabated once we resume the lockdowns? Institutions need to endorse these interventions and speed up reforms more seriously [131,132], together with endorsing collateral and more comprehensive measures [133] playing a role in epidemics and zoonoses [134,135], such as impeding biodiversity loss and land use change [136–140], decreasing intensive livestock farming, and alleviating poverty [141,142]. This new coronavirus shall be an opportunity given to the governments to forcefully revive sustainable development goals.

**Author Contributions:** Conceptualisation, R.P.; Data curation, D.F.; Formal analysis, D.F.; Investigation, R.P. and D.F.; Methodology, R.P. and D.F.; Project administration, R.P.; Resources, D.F.; Software, D.F.; Supervision, R.P.; Validation, D.F.; Visualisation, D.F.; Writing—original draft, R.P. and D.F. All authors have read and agreed to the published version of the manuscript.

**Funding:** This research received no external funding.

**Data Availability Statement:** The data presented in this study are openly available on the authors' Github repository: https://github.com/DavideFornacca/COVID19/tree/master/8_countries.

**Acknowledgments:** Lei Shi and Wen Xiao commented on the study. Livia Ottisova improved and revised the manuscript. Chun Chen and Michele Mignini commented on statistics. Rafael Moreno Ripoll and Mehrdad Samavati helped to obtain the data for Spain and Iran.

**Conflicts of Interest:** The authors declare no conflict of interest.

## Appendix A

**Table A1.** Descriptive statistics for the air pollution variables in the eight analysed countries.

| | Variable | Unit | Count | Mean | std | min | 25% | 50% | 75% | Max | Range | iqr |
|---|---|---|---|---|---|---|---|---|---|---|---|---|
| | PM25_sat | ug/m3 | 347 | 30.31 | 15.80 | 2.08 | 19.11 | 29.09 | 40.01 | 70.98 | 68.90 | 20.89 |
| | NO2_sat | ppb | 347 | 1.96 | 1.96 | 0.06 | 0.54 | 1.26 | 3.06 | 13.75 | 13.69 | 2.52 |
| | PM25_gr | AQI | 308 | 110.78 | 27.79 | 38.20 | 92.63 | 111.55 | 128.51 | 186.96 | 148.76 | 35.88 |
| China | PM10_gr | AQI | 308 | 63.61 | 22.98 | 19.96 | 46.26 | 60.90 | 75.67 | 170.27 | 150.31 | 29.40 |
| | CO_gr | AQI | 308 | 9.71 | 3.97 | 2.39 | 7.11 | 8.74 | 11.85 | 27.01 | 24.63 | 4.74 |
| | NO2_gr | AQI | 308 | 13.72 | 5.40 | 3.17 | 9.63 | 13.40 | 17.41 | 28.57 | 25.40 | 7.78 |
| | O3_gr | AQI | 308 | 25.83 | 5.87 | 14.22 | 21.70 | 25.17 | 28.98 | 50.54 | 36.31 | 7.28 |
| | SO2_gr | AQI | 308 | 13.24 | 7.96 | 1.14 | 7.88 | 10.97 | 16.46 | 40.58 | 39.44 | 8.58 |
| | PM25_sat | ug/m3 | 3104 | 9.37 | 2.68 | 2.32 | 7.26 | 9.68 | 11.54 | 15.57 | 13.24 | 4.28 |
| | NO2_sat | ppb | 3103 | 1.59 | 1.25 | 0.17 | 0.73 | 1.20 | 2.13 | 14.97 | 14.80 | 1.40 |
| | PM25_gr | ug/m3 | 429 | 7.22 | 2.10 | 0.00 | 5.88 | 7.37 | 8.66 | 15.73 | 15.73 | 2.78 |
| US | PM10_gr | ug/m3 | 203 | 16.09 | 6.36 | 4.60 | 12.41 | 15.43 | 18.62 | 40.64 | 36.04 | 6.21 |
| | CO_gr | ppm | 158 | 0.25 | 0.10 | 0.04 | 0.19 | 0.25 | 0.30 | 0.82 | 0.78 | 0.12 |
| | NO2_gr | ppb | 248 | 14.63 | 7.93 | 1.08 | 7.88 | 14.50 | 20.65 | 36.73 | 35.65 | 12.76 |
| | O3_gr | ppm | 751 | 0.05 | 0.00 | 0.03 | 0.04 | 0.05 | 0.05 | 0.06 | 0.03 | 0.00 |
| | SO2_gr | ppb | 316 | 2.61 | 5.56 | -0.38 | 0.57 | 1.30 | 2.51 | 75.47 | 75.85 | 1.94 |
| | PM25_sat | ug/m3 | 107 | 12.82 | 6.08 | 4.50 | 7.68 | 11.62 | 18.10 | 25.37 | 20.86 | 10.42 |
| Italy | NO2_sat | ppb | 107 | 2.67 | 2.42 | 0.39 | 0.99 | 1.61 | 3.95 | 11.56 | 11.16 | 2.96 |
| (provinces) | PM25_gr | ug/m3 | 90 | 16.37 | 4.89 | 6.00 | 13.00 | 15.37 | 19.46 | 29.00 | 23.00 | 6.46 |
| | PM10_gr | ug/m3 | 101 | 23.67 | 5.61 | 13.67 | 19.75 | 22.50 | 26.75 | 41.00 | 27.33 | 7.00 |
| | PM25_sat | ug/m3 | 21 | 11.29 | 4.73 | 4.90 | 7.73 | 10.48 | 13.84 | 20.59 | 15.69 | 6.11 |
| Italy | NO2_sat | ppb | 21 | 1.93 | 1.68 | 0.48 | 0.83 | 1.35 | 2.25 | 7.08 | 6.59 | 1.42 |
| (regions) | PM25_gr | ug/m3 | 19 | 15.19 | 3.50 | 9.50 | 13.00 | 14.67 | 16.05 | 22.96 | 13.46 | 3.05 |
| | PM10_gr | ug/m3 | 21 | 22.15 | 4.40 | 15.60 | 20.40 | 21.29 | 22.92 | 34.67 | 19.07 | 2.52 |
| Iran | PM25_sat | ug/m3 | 31 | 10.97 | 3.07 | 5.89 | 8.74 | 10.47 | 13.74 | 16.43 | 10.54 | 5.00 |
| | NO2_sat | ppb | 31 | 0.47 | 0.47 | 0.10 | 0.22 | 0.31 | 0.42 | 2.24 | 2.14 | 0.21 |
| France | PM25_sat | ug/m3 | 96 | 10.23 | 2.42 | 6.22 | 8.13 | 9.90 | 11.79 | 16.49 | 10.27 | 3.66 |
| | NO2_sat | ppb | 96 | 2.37 | 1.75 | 0.55 | 1.20 | 1.80 | 3.02 | 8.79 | 8.24 | 1.82 |
| Spain | PM25_sat | ug/m3 | 43 | 7.05 | 1.13 | 2.36 | 6.49 | 6.93 | 7.54 | 10.10 | 7.74 | 1.05 |
| | NO2_sat | ppb | 43 | 1.02 | 0.45 | 0.10 | 0.69 | 0.91 | 1.27 | 2.69 | 2.58 | 0.58 |
| Germany | PM25_sat | ug/m3 | 401 | 13.95 | 1.53 | 8.91 | 13.10 | 13.86 | 14.93 | 18.47 | 9.56 | 1.83 |
| | NO2_sat | ppb | 401 | 4.77 | 2.04 | 1.48 | 3.48 | 4.19 | 5.24 | 11.86 | 10.38 | 1.76 |
| UK | PM25_sat | ug/m3 | 364 | 10.51 | 2.64 | 2.16 | 9.06 | 11.29 | 12.07 | 15.37 | 13.21 | 3.01 |
| | NO2_sat | ppb | 362 | 5.47 | 2.24 | 0.48 | 4.12 | 5.26 | 6.62 | 10.23 | 9.76 | 2.50 |

## Appendix B

**Table A2.** Correlation coefficients between population density and air pollution variables in the eight analysed countries. Significant correlations (*p*-value < 0.05) are shown in bold; blue and red colour highlights indicate positive and negative correlations, respectively.

| | China | | | US | | | Italy (Provinces) | | | Italy (Regions) | | |
|---|---|---|---|---|---|---|---|---|---|---|---|---|
| | df (N-2) | Tau | *P* Value | df (N-2) | Tau | *P* Value | df (N-2) | Tau | *P* Value | df (N-2) | Tau | *P* Value |
| **PM25_sat** | 345 | 0.58 | **<0.001** | 3102 | 0.38 | **<0.001** | 105 | 0.27 | **<0.001** | 19 | 0.30 | 0.065 |
| **NO2_sat** | 345 | 0.63 | **<0.001** | 3101 | 0.54 | **<0.001** | 105 | 0.36 | **<0.001** | 19 | 0.42 | **0.007** |
| **PM25_gr** | 306 | 0.34 | **<0.001** | 427 | 0.33 | **<0.001** | 88 | 0.31 | **<0.001** | 17 | 0.45 | **0.008** |
| **PM10_gr** | 306 | 0.21 | **<0.001** | 201 | 0.27 | **<0.001** | 99 | 0.38 | **<0.001** | 19 | 0.69 | **<0.001** |
| **CO_gr** | 306 | 0.09 | **0.022** | 156 | 0.40 | **<0.001** | | | | | | |
| **NO2_gr** | 306 | 0.37 | **<0.001** | 246 | 0.52 | **<0.001** | | | | | | |
| **O3_gr** | 306 | 0.13 | **<0.001** | 749 | 0.01 | 0.824 | | | | | | |
| **SO2_gr** | 306 | 0.15 | **<0.001** | 314 | −0.09 | **0.016** | | | | | | |

| | Iran | | | France | | | Spain | | | Germany | | | UK | | |
|---|---|---|---|---|---|---|---|---|---|---|---|---|---|---|---|
| | df (N-2) | Tau | *P* Value | df (N-2) | Tau | *P* Value | df (N-2) | Tau | *P* Value | df (N-2) | Tau | *P* Value | df (N-2) | Tau | *P* Value |
| **PM25_sat** | 29 | 0.56 | **<0.001** | 94 | 0.34 | **<0.001** | 41 | 0.44 | **<0.001** | 399 | 0.14 | **<0.001** | 362 | 0.44 | **<0.001** |
| **NO2_sat** | 29 | 0.56 | **<0.001** | 94 | 0.44 | **<0.001** | 41 | 0.25 | **0.018** | 399 | 0.40 | **<0.001** | 360 | 0.39 | **<0.001** |

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
