# Peer review of "Early Spread of COVID-19 in the Air-Polluted Regions of Eight Severely Affected Countries"

_atmosphere, doi:10.3390/atmos12060795_

Round 1

Reviewer 1 Report

This paper investigated whether there is a correlation between long term exposure to air pollution and COVID-19 in second-order level administrations of eight countries: China, US, Italy, Iran, France, Spain, Germany, and UK. The work is meaningful, and the major contribution may include the eight countries’ study, but the methods and overall creativity is low. The authors stated that “this study was the first to investigate the correlation between COVID-19 and air pollution during the first wave of the pandemic.” However, there are so many related studies about the relationships between COVID-19 and air pollution. When I search “covid air pollution” in Google Scholar, I can get a lot of recent publications. Several examples are listed below.

  1. Travaglio, Marco, Yizhou Yu, Rebeka Popovic, Liza Selley, Nuno Santos Leal, and Luis Miguel Martins. "Links between air pollution and COVID-19 in England." Environmental Pollution 268 (2021): 115859.
  2. Wu, Xiao, Rachel C. Nethery, Benjamin M. Sabath, Danielle Braun, and Francesca Dominici. "Exposure to air pollution and COVID-19 mortality in the United States." MedRxiv (2020).
  3. Fattorini, Daniele, and Francesco Regoli. "Role of the chronic air pollution levels in the Covid-19 outbreak risk in Italy." Environmental Pollution 264 (2020): 114732.
  4. Wu, Xiao, Rachel C. Nethery, M. B. Sabath, Danielle Braun, and Francesca Dominici. "Air pollution and COVID-19 mortality in the United States: Strengths and limitations of an ecological regression analysis." Science advances 6, no. 45 (2020): eabd4049.

Some specific comments are:

  1. Introduction section: the literature review in the Introduction section is very weak. There are a lot of related studies as mentioned above. Please summarize the advantages and disadvantages of the previous studies, then propose your major contributions. If this paper only applies the correlation analysis to different countries, I do not think this paper is publishable. Furthermore, the two hypotheses proposed by the authors at the end of Introduction have been answered by the previous papers.
  2. Methods/Results section: as mentioned above, the methods (correlation analysis between COVID and air pollution) are very simple. If you publish this paper last May/June, this research (methods) may be fine. The paper needs to explore more to better understand this topic now, or it is a simple application to different countries. And most of the countries such as China, US, UK have already been studied. For example, this paper may propose methods to better compare the differences and similarities of the eight countries.
  3. Please be consistent for the terms such as cases/infections, deaths/mortality, and others for the COVID-19. The acronym: please only use the full name for the first appearance, then use the acronym after that such as US, UK (Line 98, 99, 109, 128, 137). Please carefully check the whole paper.
  4. Please add north arrows and scale bars in the maps (Figure 1&2).

Reviewer 2 Report

Study declares that it was analyzes the first wave of the infection. Methodology section lacks information about time frames of the first wave of COVID-19 infection in each country.

Section 2.1 should be reviewed and major changes has to be made. Pollutants are only for China, Italy and U.S.A in the methodology section. What about other countries, what pollutants were selected? What sources were used to collect air pollutants data for Iran, Spain, Germany, U.K. and France? e.g. it is visible that authors analyzed PMs and NO2 for Italy in Appendix A, Table A1.

Urban and rural areas for population density and air pollution analysis were not considered. Results section needs additional scientifically support from other scientific studies.

The sources in the figures are not indicated.

Supporting explanation why pollution level were low in selected countries could be included.

Line 265: The study does not focus on specific urban / areas " It provides some evidence that the new coronavirus infections are most often found in highly polluted and densely populated areas".

Round 2

Reviewer 1 Report

The revised version basically has no changes or improvements. I do not think it is publishable without further work.

Author Response

Dear Reviewer 1,

We are thankful for your suggestion to improve the manuscript and the time spend reviewing it.

We are sorry to hear that you are not satisfied with the answers provided. We tried to explain the history of our work. It is indeed an early analysis of the correlation between chronic air pollution and the incidence of COVID-19 performed one year ago. It was made available to the public as a preprint, but was never published in a peer-reviewed journal. We believe the publication is an officialization and a reiteration of the air pollution problem that still persists and remains underestimated (if not ignored). For this matter we provide a remarkable wealth of literature and a comparison of our findings with those of other studies in the same and additional countries.

Our best regards,

The authors

Reviewer 2 Report

Thank you for the response to comments and corrections in the draft. However, supporting results from previous studies could be presented in the results sections and could confirm / deny authors' findings.

Author Response

Dear Reviewer 2,

We are very thankful for your suggestion to improve the manuscript and the time spend reviewing it.

According to Atmosphere guidelines (https://www.mdpi.com/journal/atmosphere/instructions#preparation), the interpretation of the results in perspective of previous studies should be included in the Discussion session, not in the Results. In our MS, this can be found from line 309 to 351 for each of the countries analysed. Some additional perspectives from studies in other countries were indicated (395-403).

At the end of the Result section, we nevertheless added a new subsection with a table (Table 5) in which we have listed the countries where the correlation we have studied was also taken into account by other authors. This table reports the type of pollutants that were analysed, whether these measurements were taken from ground stations or from satellites, whether the correlation is present (positive) or absent, and, finally, whether other cofactors were concurrently analysed.

By doing so, there’s a drawback. The correlational results are further marked as showing a positive association, since in a table format it becomes more apparent that other 10 authors find this same result. We have tried to play this down since we actually don’t find a statistical significance for Germany and Spain, and we specify this in the text (lines 284-286).

We trust you are now satisfied with the improvements.

Best regards,

The authors